# Urban Flora Riches: Unraveling Metabolic Variation Along Altitudinal Gradients in Two Spontaneous Plant Species

**DOI:** 10.3390/plants13050657

**Published:** 2024-02-27

**Authors:** Elena Daniela Mogîldea, Monica Elena Mitoi, Claudia Biță-Nicolae, Dumitru Murariu

**Affiliations:** 1Institute of Biology Bucharest, Romanian Academy, 296 Spl. Independentei, 060031 Bucharest, Romania; daniela.sincu@ibiol.ro (E.D.M.); dumitru.murariu@acad.ro (D.M.); 2Faculty of Biology, University of Bucharest, 91-95 Spl. Independentei, 050095 Bucharest, Romania

**Keywords:** urban green spaces, altitudinal gradient, UV radiation, secondary metabolites, UV-absorbing compounds, *Linaria vulgaris*, *Cichorium intybus*

## Abstract

Using resilient, self-sustaining plants in urban green spaces enhances environmental and cultural benefits and reduces management costs. We assessed two spontaneous plant species, *Linaria vulgaris* Mill. and *Cichorium intybus* L., in four sites from the surrounding urban areas, ranging in altitude from 78 to 1040 m. Protection against UV-B radiation is crucial for plants at higher altitudes, guiding our focus on UV-visible absorption spectra, fluorometric emission spectra, secondary metabolite accumulation, and pigment dynamics in leaves. Our findings revealed a slight increase in UV-absorbing compounds with altitude and species-specific changes in visible spectra. The UV-emission of fluorochromes decreased, while red emission increased with altitude but only in chicory. Polyphenols and flavonoids showed a slight upward trend with altitude. Divergent trends were observed in condensed tannin accumulation, with *L. vulgaris* decreasing and *C. intybus* increasing with altitude. Additionally, chicory leaves from higher altitudes (792 and 1040 m) contained significantly lower triterpene concentrations. In *L. vulgaris*, chlorophyll pigments and carotenoids varied with sites, contrasting with UV absorbance variations. For *C. intybus*, pigment variation was similar to absorbance changes in the UV and VIS range, except at the highest altitude. These observations provide valuable insights into species-specific strategies for adapting to diverse environmental contexts.

## 1. Introduction

Selecting the appropriate plant species for urban green spaces poses a significant challenge. Opting for resilient, self-sustaining plants with high aesthetic value and pollinator-friendly flowers proves to be an excellent strategy for preserving biological diversity and enhancing the recreational benefits of these areas.

When considering urban sites at various altitudes, it is crucial to understand the strategies employed by plants to adapt to diverse environmental conditions. Several ecological factors influence plants along the altitudinal gradient, including ultraviolet (UV) radiation, soil composition, humidity, and temperature [1].

The incident UV radiation increases with altitude. For every 1000 m of altitude, the solar radiation increases by 19% in UV-B radiation, 11% in UV-A radiation, and 9% for global radiation [2]. Its level depends on the moment in the day and the atmospheric conditions. In addition to the level of UV-B radiation, an essential aspect of plant development is the ratio between UV-A radiation, UV-B radiation, and photosynthetically active radiation (PAR) [3,4]. Considering these aspects, we can say that the altitudinal gradient allows us to study plant responses to natural variations in solar ultraviolet radiation [5,6]. The depletion of the ozone layer has caused an increase in UV radiation at ground level. Even though UV-B radiation is only a small part of the total solar spectrum [7], it has a much more pronounced influence on living organisms due to the higher energy of photons (3.94∓4.43 eV) [8].

Although plants respond with quite a large variability to the action of UV-B radiation, it is generally recognized that this radiation negatively influences their growth and development. Because of their immobility, plants have developed strategies to protect themselves from the harmful action of UV-B radiation. Many studies point out the accumulation of secondary metabolites such as polyphenols, flavonoids, hydroxycinnamic acids, and tannins [9,10,11]. Around 90% of the incident UV-B radiation is attenuated before reaching the mesophyll [12]. The UV radiation in the leaf is 5–10% reflected [12,13], 2–5% is transmitted [13], and most is absorbed. Thus, absorption of UV-B radiation is an important plant protection mechanism against UV-B, especially for plants with glabrous leaf surfaces [13]. Plants growing under high UV-B radiation tend to accumulate more elevated amounts of UV-B-absorbing compounds [14,15,16]. The UV-B-absorbing compounds that plants accumulate in response to UV radiation are mainly phenolic compounds, particularly flavonoids and hydroxycinnamate esters [17]. UV-absorbing compounds play a key role in DNA protection [18] and the regulation of processes such as the inhibition of stem and leaf elongation, cotyledon expansion, and stomatal opening [19]. Among the protective roles, these compounds interfere with plant–insect relationships [20].

Chemical compounds displaying fluorescence are commonly referred to as fluorochromes [21]. Fluorochromes are generally characterized by the presence of multiple aromatic groups or planar/cyclic molecules with double bonds [22] that absorb light energy at a specific wavelength and re-emit light at a longer wavelength [23]. For example, these compounds can absorb UV radiation and emit UV and/or visible radiation. The light emission of fluorochromes can be considered biocommunication signals between plants and insects [24]. The absorption and emission capacities of plants influence the behavior of insects. Most insects have several photoreceptors in their eyes, with UV, blue, and green sensitivity peaks [25]. Tannins are phenolic compounds that have the property to absorb the UV between 200 and 282 nm [26] and have a role in the defense of plants against phytophagous insects. Tannins are some of the most abundant compounds in plant leaves, representing between 5% and 10% of their dry weight [27]. Condensed tannins are produced in different concentrations depending on the species, the developmental stage of the plant, or abiotic factors such as high temperature and low humidity [27,28,29]. Very scarce data show how these metabolites vary with UV radiation or altitudinal gradient.

UV radiation also plays a role in the synthesis of other metabolites, such as terpenes [30], that harm phytophagous insects but can attract pollinating insects [31,32,33]. Also, the UV radiation has an influence on photosynthetic pigments in plants. UV triggers distinctive alterations in the carotenoid profile and induces variations in chlorophyll levels [34,35].

The *L. vulgaris* and *C. intybus* species were selected due to their resistance, ubiquitous spread nature, ornamental value, and pollinator-friendly flowers. Additionally, both species can synthesize various substances with beneficial effects on health. These species can be found at altitudes both below and above 1000 m. However, there are limited studies addressing the altitudinal effects on these species.

Species nomenclature conforms to the Euro+Med PlantBase online database, which provides up-to-date information on vascular plants in Europe and the Mediterranean region [36].

*L. vulgaris*, known as common toadflax, is a perennial plant that is part of the Plantaginaceae [37]. It is distributed in the Eurasian region but may be present in Australia, America, South Africa, and New Zealand [38]. It is a plant that grows in cultivated or uncultivated areas, often in degraded areas [39,40]. In Romania, it grows from the plains to the mountains (BUCM Herbarium of the Institute of Biology Bucharest, registration number 131118 and 118423).

*C. intybus* is an herbaceous, ruderal plant part of the Asteraceae [37]. This plant can generally be found in Europe, West Asia, and North Africa [41,42]. It is common throughout the country in Romania, from the plains to the mountains (BUCM Herbarium of the Institute of Biology Bucharest, registration number 117251 and 110273). It can be found in pastures, ruderal places, and cultivated places [40].

This paper aims to bring new information about UV-VIS radiation absorption, secondary metabolite accumulation, and photosynthetic pigment content in *L. vulgaris* and *C. intybus* leaves along an altitudinal gradient in four sites from 78 m to 1040 m altitude (a.s.l.). The increased UV solar radiation, especially UV-B radiation, is among the first factors incriminated in plant adaptation to higher altitudes. For this reason, we were also interested in the existence of UV-absorbing fluorochromes and how their fluorescence changes across the altitudinal gradient. Because there is scarce information in the literature about the response of ruderal plants to higher altitudes, this study can provide interesting insights about adaptation to the different environmental conditions of these plants, which are commonly found at various altitudes.

## 2. Results

### 2.1. Analysis of UV and Visible Absorption Spectra

The UV-visible absorption spectra between 260 and 750 nm showed different absorption peaks, characteristic of the two studied species. Although the number of determined peaks was the same (seven), there were differences between the maximum absorption intervals in which peaks were recorded for each species (Table 1). The peaks and the absorption spectra in the UV and especially the visible range differed for these species (Figure 1a,b). The *L. vulgaris* samples had characteristic peaks in the UV-B range, while *C. intybus* had two specific peaks in UV-A (Table 1) and some supposed peaks in the UV-C range (Figure 1b). In the visible range, the absorbance values were lower in *L. vulgaris* than in *C. intybus* (Figure 1). For example, the higher values of peak 5 from *C. intybus* probably comprised peak 6, whereas for *L. vulgaris*, where the absorbance values for peak 5 were lower, peak 6 was distinguished (Figure 1a,b).

The highest absorbances to *L. vulgaris* were recorded in the UV range (Figure 1a). Two peaks were determined in the UV range: peak 1, with an absorbance maximum between 283 and 293 nm, and peak 2, with an absorbance maximum between 316 and 325 nm (Table 1). These absorbance maxima correspond to the UV-B range for peak 1 and the UV-A range for peak 2, respectively. Significantly increased values of absorbances in the UV spectrum were recorded at higher altitudes, 792 and 1040 m, compared to sites from low altitudes, 78 and 330 m (Figure 2).

In this species, five absorption peaks were discriminated in the visible range (Table 1). The higher absorbance values for the main peaks 5, 6, 7, and 10 were determined in samples from 330 m altitude (Figure 3). The absorbances of peaks 7 and 10 tended to decrease with the increase in altitude, except for the samples from the site at 330 m altitude (Figure 3c,d).

In the *C. intybus* samples, increased absorbances were observed in the visible spectra. In the UV spectra, two peaks belonging to the UV-A domain were determined (Table 1). Peak 3, with maximum absorption between 333 and 342 nm, and peak 4, with maximum absorption between 378 and 381 nm, were highlighted (Figure 1b). The high absorbance values after 280 nm wavelength and the decreasing trend suggest the presence of absorbance peaks in UV-C. The sites with higher altitudes of 330 m, 798 m, and 1040 m registered higher absorbance values for main UV and visible peaks (Figure 4). Thus, statistically significant increased values were obtained in the 1040 m site for peak 3 (Figure 4a); in the 330, 792, and 1040 m sites for peak 4 (Figure 5b); and in the 792 and 1040 m sites for peak 10 (Figure 4c).

### 2.2. Analysis of UV-B and UV-A Absorption Spectra

Other researchers consider 305 nm and 360 nm indicator wavelengths for UV-B and UV-A radiation absorption, respectively [43]. To demonstrate that these two wavelengths can be indicators for the absorption of UV-B and UV-A radiation in our species, the mean absorbances at these two indicator wavelengths were compared to the mean absorbances of the main peaks identified in our samples and the mean absorbances of the entire wavelength ranges specific to UV-B (280–315 nm) and UV-A (316–400 nm).

The absorbance at 305 nm in the species *L. vulgaris* was not different from the absorbance for peak 1 nor the absorbance for all the UV-B range (Figure 5). A grouping of absorbance values for both peak 1 and peak 2 with indicator values from 305 nm and with mean absorbances from all the UV-B range were observed. Also, the average of the absorbance values in the range of 316–325 nm (peak 2) differs statistically significantly (*p* ≤ 0.05) from the indicator values at 360 nm and from those in the range of 316–400 nm (UV-A). Peak 2 was close to the UV-B range, being recorded in the transition range from UV-B to UV-A, and can be classified as a UV-B peak.

In the case of *C. intybus* samples, no absorption peak was identified in the UV-B range. In the UV-A range, only peak 3 was considered because peak 4 was difficult to distinguish (Figure 1b). The absorbance values for all analyzed parameters were tightly grouped, with the differences being insignificant (Figure 5b).

Also, the ratio of the absorbances between the two indicator wavelengths (305/360 nm) was calculated. In *L. vulgaris* samples, absorbance at a wavelength of 305 nm was approximately twice as high as that recorded at 360 nm, whereas for *C. intybus* this ratio was around 1. In the case of the *C. intybus* species, a decrease in the subunit UV-B/UV-A ratio was observed at higher altitudes, whereas for *L. vulgaris*, an increase of more than twice this ratio was determined at higher altitudes (Table 2).

The OLS analysis indicated a weak positive correlation between the UV absorbance (280–400 nm) of the samples and the altitudinal gradient (Figure 6). According to this analysis, a 1000 m increase in altitude caused an increase in UV absorbance of 0.10 nm for *L*. *vulgaris* leaf extracts and of 0.15 nm for *C. intybus* extracts.

### 2.3. Analysis of Fluorometric Spectra with Excitation in UV-B

The properly diluted methanolic extracts from the two studied species were excited with UV radiation with wavelengths between 260 and 350 nm covering the entire UV-B range, and the emission spectra were recorded in the UV-visible range of 270–750 nm (Figure 7). In both species, the fluorochrome presence was highlighted. The higher fluorescence intensity was recorded in the UV-A range, with an emission maximum between 320 and 340 nm (peak 1), followed by an emission peak in blue between 430 and 440 nm (peak 2) and one in red between 660 and 670 nm (peak 3).

The fluorescence of the methanol extracts of *L. vulgaris* and *C. intybus* tended to increase in the visible range i.e., at peak 2 and peak 3, and to decrease in the UV range at peak 1 with increasing wavelengths of excitation radiation from UV-C to UV-B. When the excitation radiation was in the UV-B–UV-A transition range, the recorded emissions decreased for peaks 1 and 2, whereas peak 3 remained increased (Figure 7).

In the UV emission range, in both studied species, when the wavelength of the excitation radiation increased, the wavelength at which the emission peaks were registered increased. A shift to the left of λmax emission was observed. The emission at 320–340 nm (UV-A) decreased fast with each 10 nm increase in the wavelength of the excitation radiation, reaching the lowest values for excitation radiation with wavelengths above 320 nm (Appendix A). Emission peak 1 achieved the highest values at the 78 m site and decreased with increasing altitudinal gradient only in *C. intybus* (Appendix A).

In the visible range, emission peak 2 had the highest values when the excitation radiation was 310 or 320 nm for both studied species (Appendix A). At higher excitation radiation between 330 and 350 nm, the emission decreased or remained approximately at the same values. The emission values for this peak 2 did not respect an altitudinal gradient in any species. The highest emission values were recorded at the 330 m site for *L. vulgaris* and the 78 m site for *C. intybus* (Appendix A).

In both species, the emission peak 3 value increased with the increase in the wavelength of the excitation radiation. The emission values for peak 3, from the visible range, were correlated with the increase in altitudinal gradient but only in *C. intybus* (Appendix A).

### 2.4. Accumulation of Secondary Metabolites on Altitudinal Gradient

#### 2.4.1. Total Polyphenol Content

The total polyphenol concentration in the samples collected at various altitude levels did not showed an elevational gradient in any species (Figure 8a).

Higher concentrations of polyphenols were observed in toadflax than in chicory leaves. The high polyphenol content can be associated with increased UV absorption in this species. Although there was a slight increase in the polyphenolic content at altitudes higher than 330 m, the maximum recorded of 128.96 ± 2.96 μg GAE/g d.w was in the 330 m site. The Pearson correlation test showed a weak correlation of total polyphenols with altitude (r = 0.57964) (Appendix A). In chicory samples, the concentration of polyphenols increased with altitude, except for the site at 792 m, with no significant differences.

#### 2.4.2. Total Flavonoid Content

In the case of the flavonoid concentration, a slight increase in altitude was observed in both species except for the site from 792 m altitude (Figure 8b). At this site, a significantly increased value of 6.97 ± 0.29 μg RE/g d.w. (*p* ≤ 0.05) was recorded for *L. vulgaris*, whereas for *C. intybus*, a significantly lower value of 2.62 ± 0.56 μg RE/g d.w. (*p* ≤ 0.05) was determined. At the site, other ecological factors unrelated to altitude probably disturb the species-specific accumulation of flavonoids. In samples of *L. vulgaris*, the Pearson correlation test showed a weak positive correlation with altitude (r = 0.62289) and clear sky global irradiation (r = 0.52409), a positive correlation with precipitation (r = 0.77462), and a negative correlation with temperature (r = −0.77688) (Appendix A).

#### 2.4.3. Total Content of Condensed Tannins

The two species exhibited contrasting responses in their accumulation of condensed tannins concerning the altitudinal gradient. In the *L. vulgaris* species, the accumulation of condensed tannins in leaves decreases with the increase in altitude, whereas in *C. intybus*, there was an increase in leaf tannin deposition to higher altitude (Figure 8c). When the sites from the low altitudes (78 m and 330 m) were compared with those from high altitudes (792 m and 1040 m), the Student *t*-test indicated a significant decrease (*p* ≤ 0.05) from 12.75 ± 0.8 to 7.28 ± 0.6 μg CE/g d.w in condensed tannin content in toadflax leaves. In the *L. vulgaris* species, the Pearson correlation test showed a negative correlation with altitude (r = −0.76075) and clear sky global irradiation (r = −0.70341), a strong negative correlation with precipitations (r = −0.82662), and a strong positive with temperature (r = 0 0.82901) (Appendix A).

In *C. intybus*, the significant increased differences (*p* ≤ 0.05) in condensed tannin concentrations between altitudinal steps from 78 m to 330/792 m and to 1040 m were noticeable. The level of condensed tannins increased from 1.61 ± 0.12 μg CE/g d.w. at the 78 m site to 5.61 ± 0.83 μg CE/g d.w. at the 1040 m site. The Pearson correlation test showed strong positive correlation with altitude (r = 0.88778) and clear sky global irradiation (r = 0.85019), a weak positive correlation with precipitation (r = 0.65565), and a weak negative correlation with temperature (r = −0.68434) (Appendix A).

#### 2.4.4. Total Triterpene Content

The two species had different variation patterns of triterpenes leaf concentration relative to the altitudinal gradient. In the toadflax samples, the concentrations of triterpenes did not differ significantly between the sites, except for those from 78 m and 330 m (Figure 8d). No variation trend with altitude was observed in this species.

In *C. intybus*, the triterpene level in the leaves was significantly decreased (*p* ≤ 0.05) at high-altitude sites (Figure 8d). When the four collection sites were categorized based on altitude criteria, with the sites at 78 m and 330 m classified as low altitude and the other two sites at 792 m and 1040 m classified as high altitude, a notable decrease (*p* ≤ 0.05) in triterpene concentration from 1.40 ± 0.08 to 0.82 ± 0.03 μg CAE/g d.w. was observed. The Pearson correlation test showed a strong negative correlation with clear sky global irradiation (r = −0.81622) and precipitation (r = −0.79302), a weak negative correlation with altitude (r = −0.68532), and a strong positive correlation with temperature (r = 0.78939) (Appendix A).

#### 2.4.5. Determination of the Chlorophyll and Carotenoid Pigment Content

In the *L. vulgaris* species, the concentration of chlorophyll and carotenoid pigments registered the highest values at the 330 m site (Figure 9a). Variations of total pigment content were close to chlorophyll *a* (Ca) concentration, while values of carotenoid (Cx+c) concentration were grouped with those of chlorophyll *b* (Cb). Variations in these parameters at the four altitudinal sites followed the same pattern. The Pearson correlation test revealed a negative correlation of peak 1 with chlorophyll *a* (r = −0.73226), chlorophyll *b* (r = −0.75255), and carotenoids (r = −0.71203) (Appendix A).

Total pigment content in *C. intybus* showed a significant increase at the higher altitudes of 330 m, 792 m, and 1040 m compared to the 78 m site (Figure 9b). This increase would be due to the increase in the content of chlorophyll *a*, which was significantly higher in the 792 m and 1040 m sites and chlorophyll *b* in the 330 m site. The correlation between all these parameters was not maintained at the 330 and 1040 m sites. At the 330 m site, a significant increase in the concentration of chlorophyll *b* associated with a decrease in the concentration of carotenoids was noticed, while a decrease in the concentration of chlorophyll *b* and an increase of carotenoids content were observed at the 1040 m site. The Pearson correlation test revealed a strong positive correlation of chlorophyll *a* with altitude (r = 0.97), clear sky global irradiation (r = 0.94), precipitation (r = 0.96) peak 3 (r = 0.88213), UV-A range (r = 0.92673), 360 nm (r = 0.91912), and a strong negative correlation with temperature (r = −0.97) (Appendix A).

## 3. Discussion

The absorption spectra analysis in the UV and visible range suggests the presence of certain compounds at the leaf level that have the ability to absorb UV and/or visible radiation in both studied species. The differences in UV-visible absorption patterns between the two species showed that toadflax leaves possess compounds with pronounced UV absorption, particularly in the UV-B region, while chicory leaves synthesized compounds with enhanced absorption in the UV-C and visible ranges. Seven absorption peaks, corresponding to major compounds in leaf extracts, were identified in both species, but the absorption maxima were different. The absorption peaks observed in the UV range could indicate the presence of phenolic compounds, while the absorption peaks in the visible range are primarily attributed to the abundance of chlorophyll and carotenoid pigments present in the leaf extracts.

The strong positive correlation of altitude with solar radiation and precipitation and negative correlation with temperature point out that these are among the main factors that vary by altitude (Appendix A). Correlation tests indicated a dependence between UV absorbance (UV-A—peak 1, UV-B—peak 2) and altitude in *L. vulgaris* (Appendix A). Our absorbance data analysis and the strong correlation between 305 nm, 360 nm absorbance and UV-B, UV-A absorbance confirm that the wavelengths of 305 and 360 nm, proposed by Li et al. (2019) [43], can be used as specific indicators for the for UVA and UVB, respectively, in both studied species. Additionally, the multiparameter analysis of UV absorption spectra in toadflax showed that peak 2 seems to belong to the UV-B rather than the UV-A range. This peak was recorded in the transition spectrum from UV-B to UV-A and could be classified as UV-B. This implies the presence of some major compounds in the methanol extract of *L. vulgaris* that absorb UV-B, so there is specific protection against UV-B radiation in this species.

The same analyses conducted in chicory indicate closely grouped averages of absorbance recorded in UV-A, UV-B, and indicator wavelengths. This suggested the absence of differentiated protection for UV-B and UV-A radiations in *C. intybus*. The absorption peaks from UV-C or UV-A (peaks 3 and 4) probably cover the UV-B range, providing non-specific protection for UV-B and UV-A radiations.

Both species responded differently to the increase in UV radiation with the altitudinal gradient. The absorbance ratio at the two indicator wavelengths 305/360 nm suggests that the species *L. vulgaris* has a specific UV-B protection, higher than UV-A. The ratio increased at higher altitudes, indicating an enhancement in UV-B absorption with an elevational gradient. Our results confirmed the data obtained for the species *Saussurea pulchra* Lipsch., *Anaphalis lacteal* Maxim., and *Rheum pumilum* Maxim. [43].

In the case of the *C. intybus* species, a subunit decrease of 305/360 nm ratio was observed at higher altitudes, which shows an increase in UV-A absorption with altitude. This suggests an increased UVA protection with altitude in this species.

If the wavelengths at 305 and 360 nm pointed out the difference in UV-A and UV-B absorption, the results of the OLS analysis showed a weakly positive connection between all UV absorbances registered between 280 and 400 nm and altitude. Our findings support the observations of other researchers who revealed the increase in the concentration of compounds that absorb ultraviolet radiation in plants with increasing levels of UV radiation [15,16,44,45]. Also, an increase in the concentration of UV-absorbing compounds with altitude has been shown for several other plant species, such as *Polygonum chinense* L., *Hedychium gardneranum* Sheppard ex Ker Gawl., *Bocconia frutescens* L., and *Trifolium repens* L. [5].

Both species had the highest fluorescence intensity, with an emission maximum in the UV spectrum between 320 and 340 nm, followed by an emission maximum in blue between 430 and 440 nm and one in red at 660–670 nm. Maximum fluorescence in blue at 440 nm, green at 520 nm, red at 690 nm, and NIR at 720 nm was also detected in intact leaves of *Fagus sylvatica* L. [46].

Regarding the dynamics of the accumulation of some secondary metabolites at the foliar level with altitudinal gradient, our significant results were obtained only for condensed tannins in both studied species and triterpenes in chicory. Moreover, the results in the literature are contradictory. Most of the time, polyphenols and/or flavonoid classes were incriminated due to their sensitivity to increased UV radiation at high altitudes. Thus, a positive correlation between altitude and the concentration of total polyphenols was reported in *Spiraea media* L. leaves [47]. In contrast, samples collected from the lowest altitudes (1800–2000 m) had the highest values of total polyphenols in the *Seriphidium herba-album* (Asso) Sojak. than the highest altitude (2200–2600 m) [48]. An accumulation of flavonoids was detected in samples of *Fallopia japonica* (Houtt.) Ronse Decr. and *Larix kaempferi* (Lamb.) Carr. collected from high altitudes compared to those collected from low altitudes [49].

In our case, total polyphenols and flavonoid concentrations slightly increased with altitude, consistent with observations made by other researchers [47,50]. The samples from the 792 m site differed from the trend, potentially due to variations in pedological composition, elevated humidity during the collection period, or other ecological factors. Contrasting effects were observed in two experiments on *Buxus sempervirens* L., as reported by [51]. Their investigation along an altitude gradient showed an elevation-dependent increase in total phenolic acids and neolignans in leaves. Simultaneously, the concentration of flavonoids in the leaf cuticle decreased. Subsequent field experiments using UV filters did not affect these compounds significantly. This suggests that factors other than UV-B radiation contribute to the observed variations along the altitudinal gradient.

The concentration of flavonoids in our samples may be affected by precipitation and temperature (Appendix A). The temperature recorded during the growing season (March–August) was lower to higher altitudes, while the precipitation recorded the highest values at the 792 m site (Appendix A). Additionally, the high correlation between flavonoid content and the absorbance of peaks 1 and 2 suggests a predominantly flavonoid composition that provides UV protection in this species. The Pearson correlation test showed also that both UV absorbance and flavonoid concentration are dependent on altitude and solar radiation but at the same time strongly impacted positively by precipitation and negatively by temperature. These suppositions were sustained by the results of the other studies, which found that the accumulation of more flavonoids was influenced by the quantity of precipitation and registered temperatures [52].

Our data on condensed tannin accumulation with altitude were contradictory. Chicory leaves were more tanninized to higher altitudes, while tannin deposits were diminished with altitude in toadflax leaves. Another study pointed out low concentrations of tannins at high altitudes [53,54]. The accumulation of condensed tannins with altitude was reported in other studies [50,55]. Analyzing separately low-molecular-weight polyphenols (LMP) and high-molecular-weight polyphenols—tannins (HMP), other authors [56] showed that the concentration of HMP was positively correlated with altitude, unlike LMP accumulation, which was independent of altitude in the species as *Pteridium caudatum* (L.) Maxon and *Pteridium arachnoideum* (Kaulf.) Maxon.

Altitude, solar radiation, and precipitation exhibited a negative correlation with condensed tannins and a positive correlation with flavonoids, whereas temperature showed a positive correlation with condensed tannins and a negative correlation with flavonoids in *L. vulgaris* (Appendix A). These findings suggest the presence of a switching mechanism between the synthesis of condensed tannins and flavonoids in response to environmental variables. Under conditions of high photostimulation, phenol synthesis seems to be directed towards producing photoprotective compounds (flavonoids) at the expense of polymerization in the form of condensed tannins.

Our results, which showed a significant decrease in triterpenes at higher-altitude sites in *C. intybus*, align with the other observations, which noted a negative correlation between altitude and total triterpene concentration in *Cyclocarya paliurus* (Batalin) Iljinskaja leaves [57]. However, triterpene variations in our *L. vulgaris* samples cannot be correlated to the altitudinal gradient. Probably, each species has a different adaptive response to changes in environmental conditions due to higher altitudes.

Total chlorophylls and carotenoid concentrations showed variations between sites. Similar irregular variation patterns were obtained in the other species collected on altitudinal gradients, such as *Rheum pumilum* Maxim [43].

Regarding pigment accumulation in the *L. vulgaris* species, the variation patterns of chlorophylls and carotenoids between sites were opposite to the variation of absorbances in the UV range, especially peaks 1 and 2 from the UV-B range (Figure 5a and Figure 9a). Although a negative correlation existed between compounds providing photoprotection (peak 1 and 2, flavonoids) and photosynthetic pigments (Ca, Cb, Cx+c), the correlation test showed that photosynthesis was not influenced by altitude in this species. In *C. intybus*, the accumulation patterns of chlorophyll and carotenoid pigments on altitudinal gradient were similar to the absorbance variation of the peaks from the visible range (Figure 4b,c and Figure 9b). Unlike toadflax, the variation pattern of pigments was similar to the variation of absorbances in the UV range, except for the site at 1040 m (Figure 5b and Figure 9b). The higher absorbance of peak 3 was associated with increased flavonoids and condensed tannin content. Still, at the same time, there was a decrease in pigment concentration and absorbance in the visible range at the 1040 m site (Figure 9b). This suggests that the photosynthetic system of the plant was affected only after the accumulation of certain concentrations of UV-absorbing compounds, which happened to the highest monitored altitude of 1040 m. This species could be more resilient to changes in environmental conditions due to increased altitude.

In *C. intybus*, altitude is strongly positively correlated with chlorophyll *a* concentration, condensed tannins, absorbance in UV-A, and peak 3 (Appendix A). However, the very high correlation between condensed tannin content and UV-A absorbance, along with peak 3, could suggest that the majority of compounds absorbing in UV-A belong to the class of condensed tannins. Unlike *L. vulgaris*, the high correlation of biochemical variables (Ca, condensed tannin, UV-A absorbance, peak 3) with solar radiation and lower correlation with other altitude-dependent factors such as temperature and precipitation demonstrates chicory’ dependence on solar radiation, especially UV-A radiation. The close association between the variation in chlorophyll *a* concentration with altitude and other environmental factors dependent on it, as well as with UV-A and peak 3 absorbance, demonstrates that photosynthesis and UV-A protection are positively correlated with altitude in this species.

## 4. Materials and Methods

### 4.1. Characterization of the Sites Selected for Sampling

The plant samples of *L. vulgaris* and *C. intybus* from four localities distributed along an altitudinal gradient were collected (Figure 10). The vouchers of the plants are kept in the BUCA Herbarium Institute of Biology Bucharest (BUCM Herbarium of the Institute of Biology Bucharest, registration number 161835 and 161836).

The sampling sites for both species were situated at 44°22′33.71″ N, 26°3′24.07″ E in Măgurele at 78 m altitude, 45°0′12.40″ N, 25°53′1.26″ E near Băicoi at 330 m altitude, 45°19′49.90″ N, 25°33′49.56″ E in Sinaia at 792 m altitude and 45°30′29.10″ N, 25°34′38.27″ E in Predeal at 1040 m altitude. Măgurele is located in the extreme north-west of Ilfov County, in the Vlăsiei Plain and the altitude is around 78 m [58]. Băicoi is situated in the West-Southwest part of Prahova County, on the border between the Prahova Subcarpathians and the Ploieşti Plain and has an altitude between 250–406 m [59]. Sinaia is located in Valea Prahova, on the S-E slope of the Bucegi mountains and has an altitude between 798 m and 1055 m [60]. Predeal is located in Valea Prahova, in Brașov County. It has an altitude of 1030–1110 m (considered the city with the highest altitudes in the country) [61]. Sampling site map (Figure 10.) was generated using RoBioAtals, 2023 software [62].

The pedological characteristics, water regime, temperature, and solar radiation data for the study areas during the growing season (March 2022 to the end of August 2022) are detailed in Appendix A. Average monthly precipitation data and average monthly temperature data were acquired from the OGIMET database [63]. Solar radiation data are from the Copernicus Atmosphere Monitoring Service (CAMS) database [64], and UV radiation data are from the Tropospheric Emission Monitoring Internet Service (TEMIS) database [65]. The pedological characteristics data are from the soil map of Romania 1:500,000 [66].

### 4.2. Biological Material

Biological material was represented by mature leaves (1 g fresh weight) collected from five individuals located at least 4 m from each other. Since the collection areas had no arboreal layer, all the collected samples were directly exposed to the sun. Individuals chosen were mature plants showing no signs of pest attack. We avoided the leaves from the base and terminal end of the stem to reduce the risk of introducing variations caused by their degree of development. Considering the variation in the level of UV-B radiation during the day, which can influence the metabolic profile of the plants, the samples were collected in the middle of sunny days at the end of August 2022. We kept the samples cold in an isothermal bag until we returned to the laboratory.

### 4.3. Methanolic Extract Preparation

The harvested leaves were freeze-dried at −55 °C for two days at a pressure of 0.05 mbar to constant dry weight in the Alpha 1–4 LD plus lyophilizer (Christ, Berlin, Germany). The freeze-dried material was grounded with pestle mortar, and the extraction in absolute methanol 1:20 (*m*/*v*) was made for two days with continuous stirring at 280–300 rpm on a Heidolph Unimax 1010 shaker (Heidolph Instruments GmbH, Schwabach, Germany), at room temperature (RT) in the dark. The extract was centrifuged at 9000 RCF, 15 min, 4 °C using a Universal 320R centrifuge (Hettich, Kirchlengern, Germany), and the supernatant was collected.

### 4.4. Spectrophotometry and Fluorimetry Analysis

The UV-visible absorbance spectra of leaves methanolic extracts were registered in the 260–750 nm range, at an interval of 1 nm, with the Specord 210 Spectrophotometer (Analytik Jena, Jena, Germany). The methanolic extracts were diluted to 100× for common toadflax and 50× for chicory, so the recorded absorbance was around 1 unit (a.u.).

For 3-D fluorescence spectra, diluted methanolic extracts were excited with wavelengths in the UV range, between 270–350 nm, at an interval of 10 nm. The emission spectra (280–750 nm range) were registered using an FP-8300 Spectrofluorometer (Jasco, Heckmondwike, UK). The emission spectrum of the solvent was subtracted from the emission spectra of the samples.

The data were registered with WinASPECT Plus (Specord 210 Spectrophotometer software) and SpectraManager Versiunea 2 Software (FP-8300 Spectrofluorometer software) and exported and processed in Microsoft Excel 2016 [67].

### 4.5. Calculation of Chlorophyll and Carotenoid Concentrations

The same methanolic extracts were used for the determination of the content of chlorophyll and carotenoids. The concentrations of pigments were calculated in the formulas for methanolic extracts [68] with slight modifications:Ca = 16.72 A_665_ − 9.16 × A_652_(1)
Cb = 34.09A_652_ − 15.28 × A_665_(2)
Cx+c = (1000 × A_470_ − 1.63Ca − 104.96 × Cb)/221(3)
Ca+b = 1.44A_665_ + 24.93A_652_(4)

Ca—chlorophyll *a* concentration; Cb—chlorophyll *b* concentration; Cx+c—carotenoid pigments concentration; Ca+b—total chlorophyll pigments content; A665—absorbance of the samples at 665 nm; A652—absorbance of the samples at 652 nm; A470—absorbance of the samples at 470 nm.

The pigment concentrations were reported to leave dry weight and expressed in mg pigment/g d.w.

### 4.6. Determination of Total Polyphenol Content

The total polyphenol content was estimated using the standard method with the Folin-Ciocâlteu reagent (Merck, Germany) [69]. Into a sample of 0.5 mL of adequately diluted were added 2.5 mL of Folin-Ciocâlteu reagent (diluted 1:10 in AD) and 2 mL of sodium carbonate solution (7.5%). After 30 min of incubation at RT, the absorbance at 765 nm was measured against blank. All quantitative determinations were performed on the Helios Gamma UV-visible spectrometer (Thermo Fisher Scientific, Waltham, MA, USA). A standard curve (y = 0.001x + 0.0038, R^2^ = 0.9963) with gallic acid was used. The results were expressed as µg gallic acid equivalents/g dry weight (µg GAE/g d.w.).

### 4.7. Determination of Flavonoid Content

The flavonoid content was determined by adjusting the variant of the standard protocol with AlCl_3_ [70]. Thus, a mixture of 0.25 mL of diluted extract, 1 mL of AD, and 75 µL of 5% NaNO_2_ solution was incubated for 5 min at room temperature. Then, 75 µL of 10% AlCl_3_ solution was added. After 6 min at RT, the reaction was stopped with 0.5 mL of 4% NaOH, and the total volume was adjusted to 2.5 mL with AD. The samples were centrifuged for 5 min, 4 °C, at 1000 rpm, and the absorbance was measured at 510 nm against a blank. The standard curve (y = 0.0008x − 0.0044, R^2^ = 0.9993) with rutin was used. The concentration of flavonoids was expressed in µg rutin equivalents/g d.w. (µg RE/g d.w.).

### 4.8. Determination of Condensed Tannin Content

A protocol with vanillin was used for condensed tannin content estimation [71]. A sample of 0.4 mL of diluted extract was incubated with 3 mL of vanillin solution (4% in methanol) and 1.5 mL of concentrated HCl for 15 min at RT. The absorbance was red against the blank at 500 nm. A standard curve (y = 0.0018x + 0.0362, R^2^ = 0.9851) with catechin was used. The results were expressed in µg catechin equivalents/g d.w. (µg CE/g d.w.).

### 4.9. Determination of Triterpene Content

The triterpene concertation was determined using a method with acidified vanillin [72]. The diluted extract was evaporated at 70 °C, and then 0.4 mL of vanillin solution in glacial acetic acid (5%) (*w*/*v*) and 0.8 mL of perchloric acid were added on ice. The samples were incubated for 15 min at 60 °C. After cooling, another 5 mL of glacial acetic acid was added. The absorbance was measured at 545 nm against the blank. A standard corosolic acid curve (y = 0.0004x − 0.0064, R^2^ = 0.992) was used. The content of triterpenes was expressed in µg corosolic acid equivalents/g d.w. (µg CAE/g d.w.).

### 4.10. Statistical Data Analysis

The experimental data were analyzed using PAST 4.03 and 4.15 statistical software [73]. The Shapiro-Wilk W normality test was initially applied to assess the degree of data dispersion. The data were processed by ANOVA test for statistical significance assessment and the Tukey Test to determine the difference between means. For data with abnormal distribution, the Kruskal-Wallis test was used. We use the t-Student test to analyze two sets of data with a normal distribution, or the Mann-Whitney test for data with a non-normal distribution.

The OLS (ordinary least squares linear regression) model was applied to correlate the sample UV absorbance with altitude. The Pearson correlation coefficient was asses for all our studied variables and ecological factors taken in consideration.

## 5. Conclusions

Analysis of UV absorption across altitudinal gradients revealed distinct UV protection strategies among the plant species studied. In both *L. vulgaris* and *C. intybus*, slightly increased levels of UV-absorbing compounds were observed. In particular, *L. vulgaris* showed increased protection against UV-B radiation, whereas *C. intybus* did not show a differentiated defense mechanism between UV-A and UV-B radiation.

Further investigation revealed both species contained UV-B absorbing fluorochromes, with chicory showing trends in condensed tannin and triterpene accumulation. In *L. vulgaris*, flavonoids emerged as the primary contributors to UV protection, while in *C. intybus*, condensed tannins dominated in the absorption of UV-A radiation. An interesting correlation surfaced in *C. intybus*, where chlorophyll *a* concentration correlated with altitude, along with other environmental factors dependent on it, as well as UV-A and peak 3 absorbance. This illustrates a positive relationship between altitude, photosynthesis, and UV-A protection in this species.

The diverse responses observed along the altitudinal gradient in chicory and toadflax indicate the existence of varied survival or adaptation strategies to environmental conditions. These strategies enable our studied plants to overcome challenges such as increased solar radiation, lower temperatures, or enhanced precipitation conditions, which vary with altitude and other independent factors. Consequently, we consider *L. vulgaris* and *C. intybus* suitable candidates for urban green areas across sites at different altitudes.

## Figures and Tables

**Figure 1 plants-13-00657-f001:**
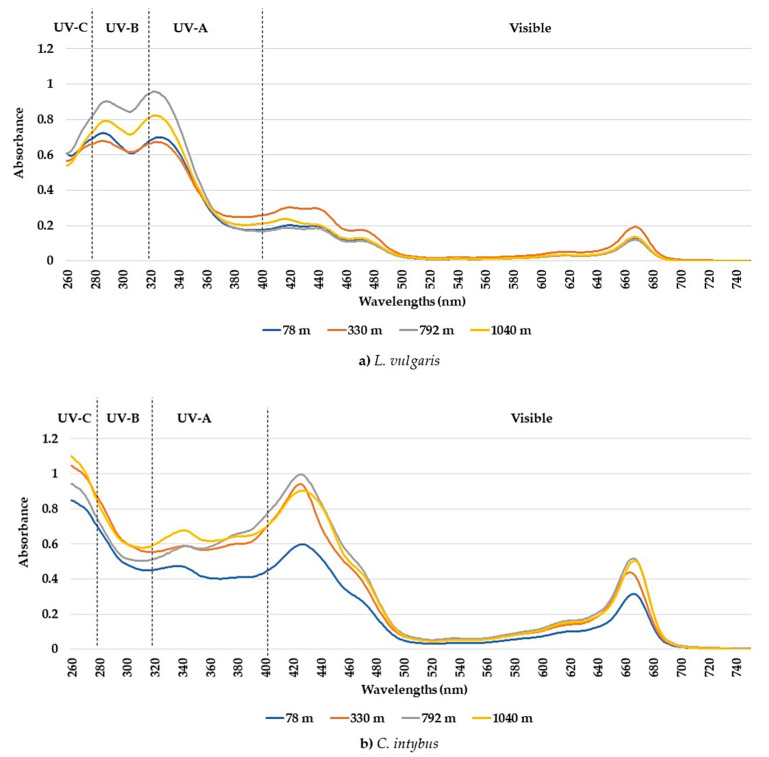
Average absorbances of methanolic extracts from leaf samples collected from the four sites at altitudes between 78 and 1040 m for the species *L. vulgaris* (**a**) and *C. intybus* (**b**).

**Figure 2 plants-13-00657-f002:**
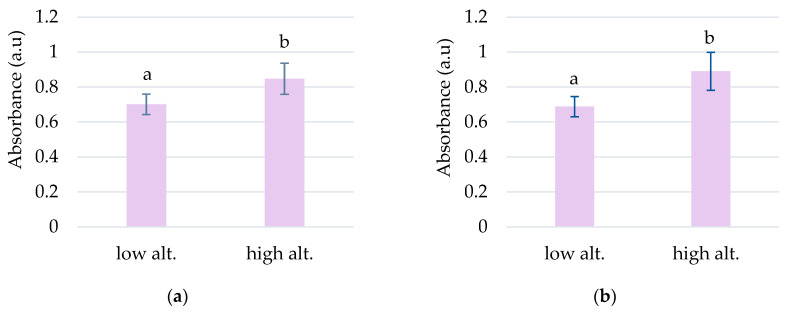
The grouping by altitude of the absorbances recorded for peak 1 (**a**) and peak 2 (**b**) to *L. vulgaris* samples. Means absorbances (M ± SD) labeled with letters significantly differ with *p* ≤ 0.001.

**Figure 3 plants-13-00657-f003:**
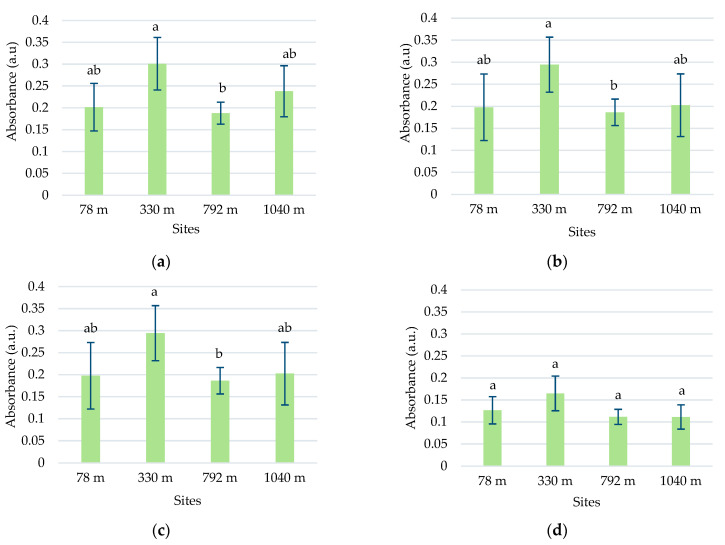
Average absorbances (M ± SD) of *L*. *vulgaris* samples, recorded in the visible range between 414 and 422 nm for peak 5 (**a**), 437 and 446 nm for peak 6 (**b**), 469 and 476 nm for peak 7, (**c**) and 666 and 672 nm for peak 10 (**d**). Means labeled with different letters are statistically significant with *p* ≤ 0.05.

**Figure 4 plants-13-00657-f004:**
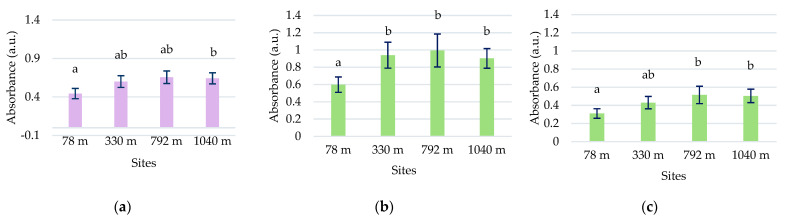
Mean absorbances (M ± SD) of *C*. *intybus* samples were recorded between 333 and 342 nm for peak 3 (**a**), 424 and 430 nm for peak 5 (**b**), and 664 and 668 nm for peak 10 (**c**). Means labeled with different letters are statistically significant with *p* ≤ 0.05.

**Figure 5 plants-13-00657-f005:**
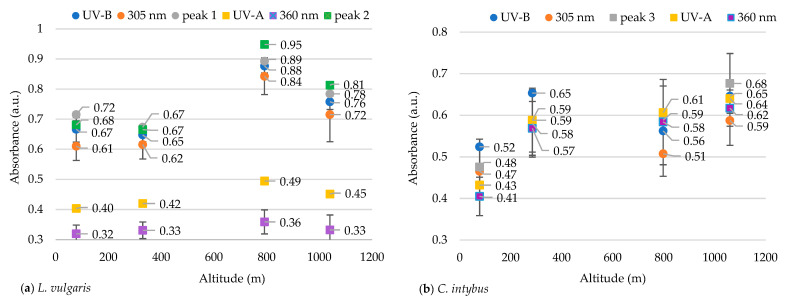
The average absorbances in the UV-B and UV-A range in the methanolic extracts of leaves of *L. vulgaris* (**a**) and *C. intybus* (**b**).

**Figure 6 plants-13-00657-f006:**
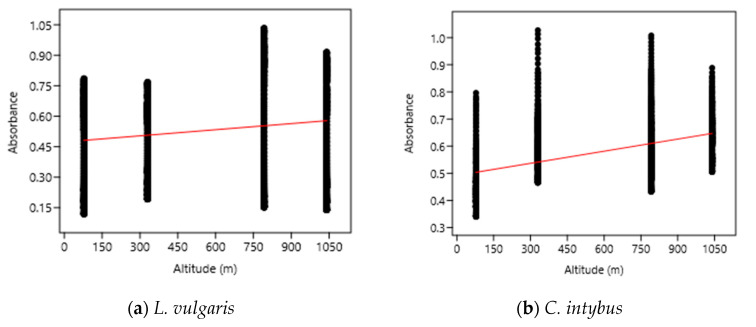
The OLS analysis between UV absorbance and altitude for *L. vulgaris* (**a**) and *C. intybus* (**b**) calculated with the linear equations y = 0.47395 + 0.00010033x and r^2^ = 0.02232 for *L. vulgaris* (**a**) and y = 0.49228 + 0.00014876x and r^2^ = 0.024405 for *C. intybus* (**b**).

**Figure 7 plants-13-00657-f007:**
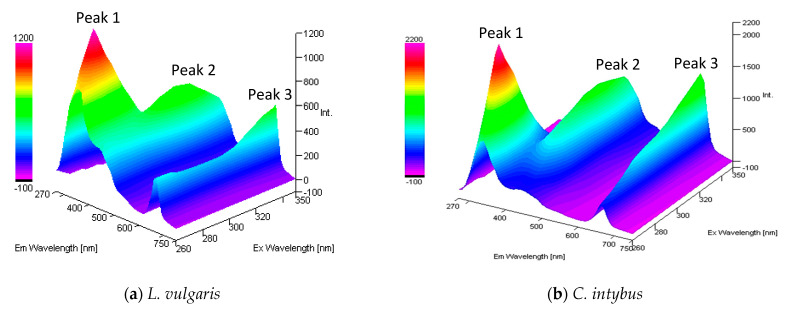
The 3-D absorption/emission spectra of the methanolic extract from *L. vulgaris* (**a**) and *C. intybus* (**b**) leaves, diluted 100 and 50×.

**Figure 8 plants-13-00657-f008:**
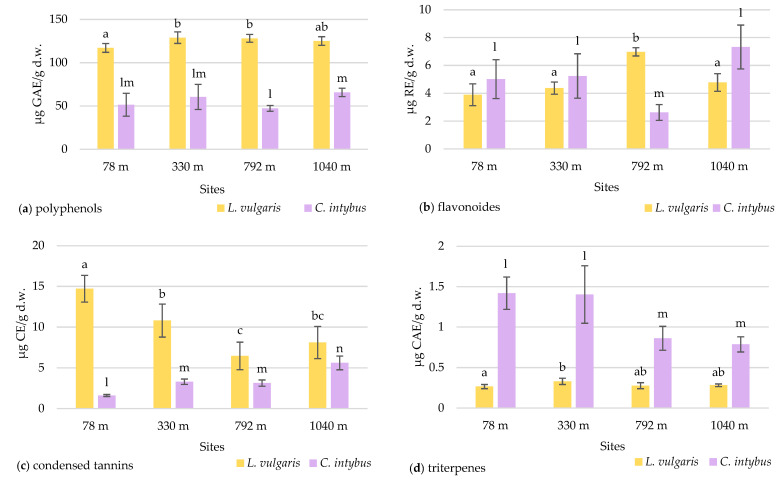
The variation in the total content of polyphenols (**a**), flavonoids (**b**), condensed tannins (**c**), and triterpenes (**d**) in *L. vulgaris* and *C. intybus* leaves collected from the four urban sites located at different altitudes. Data are presented as means and standard deviations (M ± SD). Means labeled with different letters (a, b, c for toadflax and l, m, n for chicory) are statistically significant with *p* ≤ 0.05.

**Figure 9 plants-13-00657-f009:**
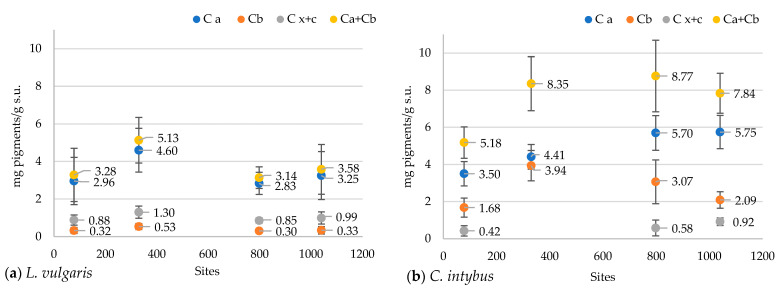
The concentrations of chlorophyll and carotenoid pigments in the methanolic extracts of *L. vulgaris* (**a**) and *C. intybus* (**b**) leaves at the four sites located at different altitudes. Data are presented as means and standard deviations (M ± SD).

**Figure 10 plants-13-00657-f010:**
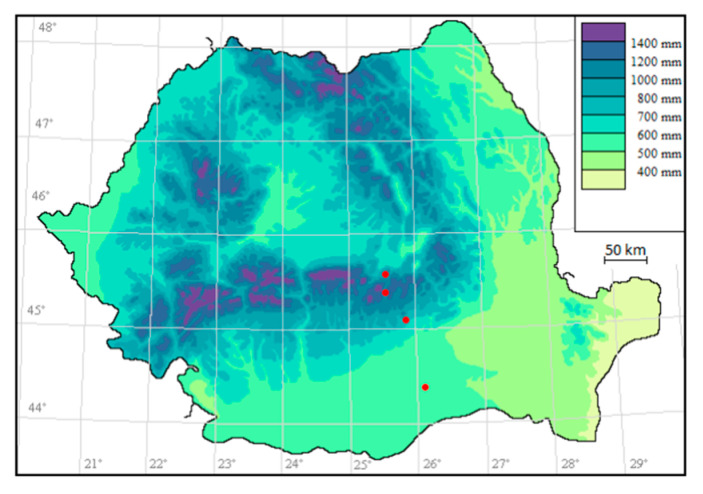
Sampling sites (represented on the map in red).

**Table 1 plants-13-00657-t001:** The wavelength ranges where absorption peaks were recorded for each species.

Wavelength Range	Peak No.	The Maximum Absorption Intervals for Each Peak (nm)	
		** *L. vulgaris* **	** *C. intybus* **
UV-B (280–320 nm)	1	283–293	
UV-A (320–400 nm)	2	316–325	
	3		333–342
	4		378–381
Visible (400–750 nm)	5	414–422	424–430
	6	437–446	
	7	469–476	464–472
	8		536–539
	9	616–618	616–620
	10	666–672	664–668

**Table 2 plants-13-00657-t002:** The ratio between the indicator absorbance at 305 and 360 nm.

Species	Altitude (m)			
	78	330	792	1040
*L. vulgaris*	1.911	1.860	2.347	2.151
*C. intybus*	1.148	1.021	0.869	0.951

## Data Availability

All data acquired during this study are incorporated into this paper as figures, tables, and Appendix A.

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
