# Peer review of "Urban Flora Riches: Unraveling Metabolic Variation Along Altitudinal Gradients in Two Spontaneous Plant Species"

_plants, 2024, doi:10.3390/plants13050657_

Round 1

Reviewer 1 Report

Comments and Suggestions for Authors

In the paper, the authors identified metabolic changes in two spontaneous plant species in four sites in adjacent urban areas at altitudes ranging from 78 to 1040 m above sea level. The effect of UV radiation on metabolic changes was mainly assessed.

This topic is relevant because it helps to understand the mechanism of plant adaptation to changing conditions.

This study, compared to others, reveals in sufficient detail the influence of natural UV radiation on secondary metabolites of important agricultural plants.

The introduction provides sufficient background and includes an overview of previous research. Materials and methods are described in sufficient detail.

I have some small comments:

Data on the intensity of UV radiation in the study areas should be added if such measurements have been carried out.

The main conclusions are consistent with the results of the study. The authors observed different trends in the accumulation of tannins, triterpenes, pigments and other secondary metabolites.   References to literature are appropriate.

Figures 6 and 13 are difficult to read. The data can be presented in the form of a diagram.

You can add quantitative measurement results to the annotation.

Check your text formatting.

Author Response

Thank you for taking the time to review this manuscript. We appreciate your valuable and helpful suggestions for revising and improving our manuscript.

Reviewer 2 Report

Comments and Suggestions for Authors

Comments:

In the work “Urban Flora Riches: Unraveling Metabolic Variation Along Altitudinal Gradients in Two Spontaneous Plant Species” (plants-2844793), the authors measured the UV-VIS spectra of methanolic extracts in two plant species, Linaria vulgaris Mill. and Cichorium intybus L., in four sites along an altitudinal gradient, the dataset may be good for the community but not the analysis. The study is interesting and falls within the journal's scope well. Unfortunately, the results are not solid for their conclusion, e.g., many observed variables are not elevation-dependent. (The reason might be the choice of the altitude range is quite small) My concerns may be useful in their revised manuscript when submitting to another journal.

Major issues:

i. The OLS analysis between UV absorbance and altitude is good, but the variation explanation is extremely low, 2.232% for L. vulgaris and 2.4405% for C. intybus, which means UV absorbance might not be influenced by the elevation.

ii. In figure 4, the VIS in L. vulgaris samples, and in figure 5, the mean absorbances of in intybus samples have tiny differences although the authors declared a significant difference.

iii. Also, the variation of the total content of polyphenols in L. (figure 9), variation of flavonoid concentration in (figure 10), variation of the concentration of condensed tannins (figure 11), and variation of the concentration of triterpenes in L. vulgaris (a) and C. intybus (b) leaves are not elevation-dependent.

iv. The analysis of the chlorophyll and carotenoid pigment contents is interesting, but what are the linkages with other observed variables in your study?

Small issues:

L37: “The most critical factors for plants are UV radiation and temperature”, I cannot agree.

L70: What is “VIS”? If it’s a new word, you should define it at its first appearance.

Author Response

Thank you for your valuable comments. We appreciate your recognition of the alignment of the study to the scope of the journal. We acknowledge your concerns about the robustness of our results and the potential limitations associated with the chosen altitude range. Underlying this study are observations and research conducted over three years as part of the first authors PhD thesis and introduce new perspectives to the field. Moreover, we bring other authors arguments to support our findings.               We are carefully considering your comments in preparation for revisions and will ensure that our revised manuscript addresses the issues raised. While we recognize the importance of your comments, we also believe that the results contribute significantly to the existing knowledge base.

Reviewer 3 Report

Comments and Suggestions for Authors

REVIEW REPORT

The work is well planned and described, the literature review is quite extensive. The work contains all the necessary elements of scientific research and after minor corrections, the results of the work can be published. However there are many many mistakes in spelling. Please read the entire work carefully again and correct them. The following types of errors are listed below:

1.      Line 117, delete coma before citation, add dot at the end of sentence.

2.      Line 138 – how many leaves?

3.      Figure 1 – add space before bracket of (1)

4.      Line 239 – add dot after Fig

5.      Line 241 – “Table” not “tabel”

6.      Figure 3 (and other places in the work) – p should be written in italics (p)

7.      Line 251 – add space between “and” and “10”

8.      Line 253 – delete space between 4 and c.

9.      Line 269 (Figure 5) – add space after p

10.   Line 303 – add space before 7

11.   Line 337 – add italics for species name (L. vulgaris)

12.   Line 350 and 369 (Figure 9 and 10) – add space after p

13.   Line 381 – add dot after sentence.

14.   Line 393 – delete space between 12 and b.

15.   Line 404 and 405 – it should be written in italics a and b for chlorophyll a and b (also correct in other places of manuscript)

16.   Line 411 – delete space between 13 and b.

17.   In literature some of positions in numbering are bolded (e.g. 67-73).

Additionally:

1.      In supplementary materials on the charts, please translate the month ranges into English (Figure S1-S3). Also there are some dots marked in “track changes” function of the word processor.

2.      In my opinion, figures like 12 or 11 (and others like these) when you compare same results but for different species should be shown on the one chart (2 different colors of bars). Or at least label the graphs in the figure with the names of the species they refer to.

Author Response

We truly value the time and effort you invested in giving feedback on our manuscript. We are grateful for your precious advice aimed at revising and enhancing our manuscript.

Reviewer 4 Report

Comments and Suggestions for Authors

The manuscript entitled: ‘Urban Flora Riches: Unraveling Metabolic Variation Along Altitudinal Gradients in Two Spontaneous Plant Species’ is a comprehensive research on the species-specific strategies for adapting to diverse environmental contexts. In my opinion, such a detailed research will certainly be of great interest to researchers around the world as the study addresses the highly important topic of plant adaption to environment. The manuscript is suitable for publication in the journal after minor revision.

The title represents well the content of the manuscript.

The English language is correct, however, I suggest to remove the minor errors, such as the abbreviation of the binomial names: line 94: should be “L. vulgaris”, line 99: should be “C. intybus”.

The methodology and results were presented in a clear manner.

The Discussion is extensive and the comparison with other findings deserves special attention. However, the conclusion is not specific enough, Authors should summarize the main findings of this paper.

Author Response

(The authors gave the same response as above.)
